# MOMFNet: A Deep Learning Approach for InSAR Phase Filtering Based on Multi-Objective Multi-Kernel Feature Extraction

**DOI:** 10.3390/s24237821

**Published:** 2024-12-06

**Authors:** Xuedong Zhang, Cheng Peng, Ziqi Li, Yaqi Zhang, Yongxuan Liu, Yong Wang

**Affiliations:** 1School of Geomatics and Urban Spatial Informatics, Beijing University of Civil Engineering and Architecture, Beijing 102616, China; zhangxuedong@bucea.edu.cn (X.Z.); 2108160122003@stu.bucea.edu.cn (C.P.); 2108160122011@stu.bucea.edu.cn (Z.L.); 2Beijing Key Laboratory of Urban Spatial Information Engineering, Beijing 100038, China; 3Beijing Institute of Surveying and Mapping, Beijing 100045, China; liuyongxuan123@126.com; 4Beijing Urban Construction Survey and Design Institute, Beijing 100101, China; wangy3029@163.com

**Keywords:** InSAR, MOMFNet, twisted 2D Gaussian surfaces, multi-kernel feature extraction, multi-objective loss function

## Abstract

Interferometric Synthetic Aperture Radar (InSAR) is a widely used remote sensing technology for Earth observation, enabling the detection and measurement of ground deformation through the generation of interferograms. However, phase noise remains a critical factor that degrades interferogram quality. To address this issue, this study proposes MOMFNet, a deep learning approach for InSAR phase filtering based on multi-objective multi-kernel feature extraction that leverages multi-objective multi-kernel feature extraction. MOMFNet incorporates a multi-objective loss function that accounts for both the spatial and statistical characteristics of the denoising results, while its multi-kernel convolutional feature extraction module captures multi-scale information comprehensively. Furthermore, the introduction of weighted residual blocks allows the model to adaptively adjust the importance of features, improving its ability to accurately identify and suppress noise. To train the MOMFNet network, we developed an interferogram simulation strategy that uses randomly distorted 2D Gaussian surfaces to simulate terrain variations, Perlin noise to model atmospheric turbulence phases, and negative Gaussian noise to generate random training samples at different noise levels. Comparative experiments with traditional denoising methods and other deep learning approaches, through both qualitative and quantitative analyses, demonstrated that MOMFNet excels in noise suppression and phase recovery, particularly in scenarios involving large gradients and random noise. Empirical studies using Sentinel-1 satellite data from the Yanzhou coal mine validated the practical value of MOMFNet, showing that it effectively removes irrelevant noise while preserving critical phase details, significantly improving interferogram quality. This research provides important insights into the application of deep learning for InSAR denoising.

## 1. Introduction

Interferometric Synthetic Aperture Radar (InSAR) is a sensing technique that uses radar images to obtain information about surface height changes. By performing interferometry on two or more synthetic aperture radar images, InSAR extracts surface height variation information and can further generate a digital elevation model (DEM) of the observed area [1,2]. However, the interferometric phase of SAR is easily affected by complex random noise, which severely impacts the accuracy of the results [3]. Therefore, conducting denoising is crucial for phase unwrapping, as it can weaken the influence of noise on phase unwrapping, thereby obtaining more reliable measurement results [4,5].

Over the past few decades, significant progress has been made in InSAR phase denoising algorithms, which can be divided into four stages: (1) Local Filtering Stage. Early methods in this stage mainly focused on local filtering by analyzing local information of pixel neighborhoods to estimate the phase. Common local filtering methods include the following. Statistical-based filtering involves estimating the phase using statistical methods such as mean and median, e.g., mean filtering and median filtering [6,7]. Local window-based filtering involves using local windows for filtering, e.g., Lee filtering and adaptive local noise filtering [8,9,10]. Local frequency-based filtering involves using polynomial models to simulate the local phase, e.g., linear or nonlinear local frequency filtering [11,12,13]. Component decomposition-based filtering involves decomposing the phase into principal and residual components and filtering them separately [14,15]. (2) Transform-Domain Filtering Stage. In this stage, besides spatial domain filtering, transform-domain filtering is also widely applied, such as Fourier Transform and Discrete Cosine Transform. Goldstein filtering and its improved versions are currently among the most typical and effective transform domain filtering methods [16,17,18,19,20,21]. Some of these methods utilize multi-resolution representation to enhance phase features and achieve denoising [22]. (3) Non-Local Filtering Stage. Non-local filtering techniques became a research hotspot in this stage, utilizing non-local similarity in images for filtering [23]. For instance, probability patch-based filtering involves using weighted maximum likelihood estimation for filtering [24]. Block matching 3D filtering (BM3D) involves adapting traditional BM3D filtering for InSAR applications [25]. (4) Sparse Signal Processing Stage. With the development of the signal processing field, sparse signal processing was introduced to InSAR phase denoising. For instance, sparse regularization involves using Bayesian rules to restore sparse representations of images through maximum a posteriori estimation [26]. Sparse coding involves improving the sparse representation of phase images through dictionary learning [27,28]. In summary, InSAR phase denoising algorithms have continuously evolved from early local filtering to transform-domain filtering, non-local filtering, and sparse signal processing, reflecting a deepening understanding of phase information and advancements in algorithm technology.

In recent years, deep learning (DL) has achieved significant results in the field of image processing and has gradually been applied to the processing workflow of InSAR. This has brought new breakthroughs in tasks such as deformation detection, interferogram denoising, phase unwrapping, and time-series subsidence prediction [29,30,31,32]. Unlike traditional InSAR processing methods based on mathematical models and assumptions, DL methods establish complex relationships between inputs and outputs by learning features from large training datasets. Specifically, DL models use training datasets to learn the mapping relationship between input data and reference results, updating parameters based on the differences between generated outputs and references, and ultimately achieving alignment between model outputs and reference results, thus improving the accuracy of the method.

In the field of InSAR denoising, DL methods vary in terms of architecture design, datasets, and cost functions. Yu et al. considered the impact of resolution loss and fringe distortion on denoising results and proposed a non-local (NL) phase denoising method, PDNet, based on deep convolutional neural networks (DCNN) [33]. Zhao et al. introduced channel and spatial attention mechanisms, emphasizing the preservation of the edge features of interferometric phases based on noise suppression, and proposed a multi-priority mechanism-based deep learning network (MAPDNet) [34]. Yang et al. proposed a wrapped phase denoising method based on a diffusion model, considering the impact of random noise on interferograms and based on new conditions for phase sine and cosine [35]. S. Mukherjee et al. proposed two new methods: an unsupervised training method based on an autoencoder for InSAR phase denoising and a framework named “GenInSAR” that combines phase filtering and coherence estimation and can directly learn the distribution of InSAR data [36]. Sun et al. proposed another CNN-based supervised solution, DeepInSAR, for jointly estimating filtered phase and coherence [37]. Li et al. proposed a network named IPDnCNN, aiming to achieve denoising by estimating phase noise and subtracting it from the noisy phase signals [38].

In summary, existing InSAR phase denoising methods typically focus on specific model architectures or the construction of simulated datasets, relying on simple loss functions to evaluate the discrepancy between the model output and the reference data.

To overcome these limitations, this study presents MOMFNet, A Deep Learning Approach for InSAR Phase Filtering Based on Multi-Objective Multi-Kernel Feature Extraction. Unlike conventional methods, MOMFNet employs multi-scale feature extraction using convolutional kernels of varying sizes and introduces a multi-objective loss function with learnable weighting parameters to achieve more comprehensive denoising performance. Specifically, MOMFNet constructs a three-part cost function aimed at optimizing global spatial features, edge details, and statistical characteristics. Additionally, the MF-SENet module is integrated for multi-scale information acquisition in the interferogram. To train MOMFNet, we employed a mathematical model to randomly generate a training dataset that includes deformation phases, atmospheric turbulence phases, and noise phases. This approach not only produces noiseless interferometric phase data but also more accurately reflects edge features, thereby enhancing the quality of the training data. To validate the effectiveness of the proposed method, MOMFNet was subjected to both quantitative and qualitative evaluations using simulated and real datasets, and its results were compared with classical phase denoising algorithms. The results demonstrate significant performance improvements with MOMFNet in phase denoising, effectively preserving edge details and statistical characteristics, highlighting its potential for application in InSAR data processing.

The organization of this study is as follows. Section 2 introduces the research data, including both simulated and real data. This is followed by an overview of the filtering methods, covering model construction and the design of multi-objective functions. Finally, the evaluation metrics and validation methods used in the experiments are presented. Section 3 discusses the experimental results, examining the model’s performance on simulated datasets and its practical applications. Section 4 provides the conclusions and future outlook.

## 2. Materials and Methods

### 2.1. Data Sources

#### 2.1.1. Simulation Data

Due to the complexity and randomness of noise in real interferograms, relying solely on sufficient ground measurement data for training is challenging and often insufficient to address practical problems. To overcome this difficulty, this study adopts a method of training with simulated interferograms that closely mimic real-world features [39].

To enhance training efficiency and to observe changes in subsidence information more clearly before and after denoising, this study chooses to construct mining area datasets. The training sample size is set to 180 × 180 pixels, covering an area of approximately 3.6 × 3.6 km. This approach not only improves the efficiency of model training and inference but also covers the vast majority of mining areas [40]. The phase of simulated interferograms consists of a deformation phase, an atmospheric turbulence phase, and a noise phase.

The deformation phase in mining areas is mainly simulated using distorted two-dimensional elliptical Gaussian surfaces, which resemble the bell-shaped mining subsidence pattern [41]. By optimizing parameters, diverse forms of subsidence information can be generated. The two-dimensional elliptical Gaussian function can be represented as Equation (1):(1)$$\left(u)=\frac{1}{2\pi {\left|N\right|}^{-\frac{1}{2}}}\exp[-\frac{1}{2}{\left(u-v\right)}^{T}N^{-1}(u-v)\right]$$
where $u=(u_{1},u_{2})$ represents the two-dimensional grid of interferograms in the training set, $v=(v_{1},v_{2})$ controls the deformation center, and the covariance matrix is defined as in Equation (2):(2)$$N=s\times U^{\prime }BU$$

Considering the influence of various factors on ground deformation, including the distribution of mining areas, the complexity of underground structures, and the characteristics of different mining methods, this study applies random deformation to Gaussian elliptical surfaces to simulate deformations under various practical scenarios, as shown in Figure 1b. Specifically, using m×n control points distributed with random matrices ($V_{x}^{m\times n}$, $V_{y}^{m\times n}$), determines the direction and magnitude of distortion on a standard two-dimensional Gaussian surface. This is detailed in Equations (3) and (4):(3)$$\delta_{x}=\alpha \cdot \sum \left(V_{x}\cdot \varphi \right)$$
(4)$$\delta_{y}=\alpha \cdot \sum \left(V_{y}\cdot \varphi \right)$$
where α represents the overall distortion level and φ denotes the distortion magnitude at each control point. The calculation method is described in Equation (5):(5)$$\varphi =R{\left(1-\frac{d}{l}\right)}^{p}$$
where
$$R=-2t^{3}+3t^{2}$$
where *l* represents the diagonal length of the standard two-dimensional Gaussian surface, used for normalizing the Euclidean distance d from a pixel to all control points; parameter *p* is introduced as an enhancement factor to adjust the dependency on distances between control points. Additionally, a smoothing function *R* is designed, with derivatives at *t* = 0 and *t* = 1 being zero, thereby avoiding discontinuities in simulating deformation signals. After applying the randomly generated parameters, the distorted deformation phase can be obtained.

This study utilizes Perlin fractal noise to model atmospheric phase delay. Perlin noise has been widely used to reproduce natural textures, such as simulating hills, flames, clouds, and rocks [42,43,44], and has also been applied to simulate atmospheric phases in SAR interferograms [45]. In this research, by combining multiple Perlin noises of different frequencies and amplitudes, the resulting noise pattern becomes more complex and natural, generating local atmospheric turbulence phases. The deformation phase and turbulence phase are combined as a true phase for training, as shown in Figure 1c. Gaussian complex noise is then added to simulate decorrelation effects. Gaussian complex noise combines the statistical properties of Gaussian distribution with the processing capabilities in the complex domain, making it a crucial noise model. Additionally, the noise level increases with the deformation gradient, allowing it to simulate real noise scenarios. The generated noise is represented as Equation (6):(6)G=ω⋅F(g)
where ω is a standard normal distribution random matrix of the same size as the training samples *g* represents the normalized phase gradient. F(g) denotes a linear mapping and is defined as Equation (7):(7)F(g)=(NL−η)⋅g+η
where NL is a parameter determining the maximum standard deviation of noise. η limits the minimum noise level and is randomly set during the simulation process, constrained by the condition 0≤η≤NL.

Finally, independent noise matrices are generated for the real and imaginary parts, with equal noise levels. These two matrices are then combined to form a complex noise matrix, which is subsequently multiplied with a simulated interferogram to generate synthetic data varying with different noise levels, as shown in Figure 1d. To better present the interferograms with different noise levels in the training set, this study sets two constant values of π and 2π, as shown in Figure 1e.

#### 2.1.2. Real Data

The research area is located in Yanzhou City, Jining City, Shandong Province, China. Its center is approximately 35°03′01″ north latitude and 115°56′18″ east longitude, with an altitude of about 47 m. We used Sentinel-1 data from the European Space Agency, with a DEM of 30 Meter SRTM (https://search.asf.alaska.edu/, accessed on 20 March 2023).

### 2.2. Filtering Method

#### 2.2.1. MOMFNet Construction

The design of MOMFNet is inspired by the DnCNN image denoising neural network. DnCNN emphasizes the synergy between residual learning and batch normalization, allowing for fast convergence and outstanding performance even in deep networks [46]. DnCNN employs a fully convolutional architecture without downsampling layers, stacking convolutional layers to build a network capable of handling input samples of arbitrary size. To further enhance the model’s generalization and representational abilities, MOMFNet extends DnCNN by introducing the MF-SENet module for multi-kernel feature extraction, as well as weighted residual blocks that improve generalization and expressive power [47]. Additionally, to improve computational efficiency, MOMFNet incorporates skip connections, enabling more efficient information flow, which further enhances the model’s performance [48]. Specifically, the network structure of MOMFNet is designed as Figure 2.

The first part of the model employs 64 convolutional filters with a kernel size of 3 × 3 to extract 64 feature maps from the interferogram. These feature maps are then processed by a ReLU activation function, introducing nonlinearity to enhance the model’s representational capacity.

The second part constructs the MF-SENet module, which consists of the MF module and SENet and incorporates skip connections between the input and output to accelerate network training and improve performance. The input data first passes through the MF module, whose core lies in using convolutional filters of three different kernel sizes (3 × 3, 5 × 5, and 7 × 7) to extract multi-scale information. Each kernel size generates a set of 64 feature maps containing information at different scales, which are then processed through batch normalization (BN) and ReLU activation before being concatenated. Batch normalization normalizes each mini-batch during training, effectively mitigating issues such as vanishing or exploding gradients and improving the network’s robustness to learning rate variations [49,50].

Subsequently, 192 filters of size 3 × 3 are used to fuse the multi-scale information, and the result is passed as input to SENet (Squeeze-and-Excitation Networks). SENet is a channel attention mechanism-based model that learns the importance of each channel and adjusts the feature maps accordingly. By multiplying the learned channel weights with the feature maps, SENet amplifies important features while suppressing less relevant ones, thereby enhancing the feature representation. The detailed process is as follows:(1)In the Squeeze stage, global average pooling is applied to compress the input feature maps into a channel descriptor, representing the global characteristics of each channel. Specifically, the output of the Squeeze stage is represented as Equation (8):
(8)$$Z=AvgPool(X)\in R^{192\times 180\times 180}$$
where *AvgPool* represents the global average pooling operation, *X* denotes the input feature map, and *R* is its shape, which in this study is 192 × 180 × 180.(2)In the Excitation stage, a series of fully connected (*FC*) layers and activation functions (such as *ReLU*) are employed to learn the weight coefficients for each channel, which are used to adjust the feature response of the corresponding channels. Specifically, the output of the Excitation stage is represented as Equation (9):
(9)S=σ(fReLU(fFC(Z))) where $f_{FC}$ represents the fully connected layer, fReLU denotes the *ReLU* activation function, and σ refers to the Sigmoid activation function.(3)In the Scale stage, the learned weight coefficients are multiplied by the corresponding feature maps to achieve the weighted operation of the feature maps. Specifically, the output of the Scale stage is represented as Equation (10):
(10)Y=X⋅S where ⋅ denotes element-wise multiplication.

The third part is the weighted residual block, which receives the feature maps from the previous stage. In traditional residual connections, the original input features are directly added to the features processed by the convolution operation. However, in the weighted residual connection, a learnable weight is introduced to perform a weighted summation of the two feature sets. This learnable weight is updated during network training, allowing the network to adaptively adjust the importance between the input features and the features processed by the convolution operation, thereby increasing the network’s flexibility. The corresponding expression is Equation (11):(11)$$D=\omega \cdot Y+(1-\omega )f_{BR}(Cov(Y))$$
where ω represents the learnable parameter, Y denotes the input features, and $f_{BR}$ refers to the features processed by batch normalization. *Cov* is the convolution operation.

In the fourth part, the model employs 3 × 3 convolutional filters to reconstruct the desired output, producing the unwrapped phase map. To further enhance the model’s feature extraction, reconstruction, and parameter optimization capabilities, the depth of both the second and third parts was extended to eight layers. This expansion aims to mitigate potential overfitting issues within the model.

#### 2.2.2. Construction of Multi-Objective Loss Function

In deep learning, the loss function measures the discrepancy between model predictions and the true labels, playing a critical role in evaluating model performance [51]. Selecting an appropriate loss function is crucial for training effective deep learning models, as it directly impacts learning outcomes. However, most studies focus on designing specific loss functions, which can lead to overly uniform patterns in model parameter learning, especially in complex stochastic tasks like phase denoising.

To address the limitations of single loss functions in existing methods, this study proposes a multi-objective loss function aimed at simultaneously preserving spatial and statistical characteristics after denoising. This multi-objective loss function combines three terms linearly, each tailored to evaluate different metrics of spatial fidelity before and after denoising, thereby guiding model parameter updates towards optimal directions.

Specifically, the first term for overall spatial reconstruction can be represented as Equation (12):(12)$$L_{1}=\frac{1}{n\times m}\sum_{i=1}^{n}\sum_{j=1}^{m}1-\cos(D_{0}(i,j)-D_{1}(i,j))$$
where n×m in this study is 180 × 180 and $D_{0}(i,j)$ and $D_{1}(i,j)$ represent the phase values of the model’s output denoised image and the true denoised image, respectively, at position (i,j).

Unlike traditional mean squared error (MSE), this study employs the cosine theorem to assess the difference between model output phase values and true phase values. This method avoids sudden discontinuities in the comparison process and reduces errors in the loss function [52].

The second term focuses on changes in edge details. Given that deformation gradients in mining areas are typically large, deformation edges are also crucial indicators for evaluating denoising results. Specifically, this is represented as Equation (13):(13)$$\left\{\begin{array}{l} A_{1}=\sum_{i=1}^{n}\left|\nabla_{u}D_{0}(i)-\nabla_{u}D_{1}(j)\right|\\ A_{2}=\sum_{i=1}^{n}\left|\nabla_{v}D_{0}(i)-\nabla_{v}D_{1}(j)\right|\\ L_{2}=\frac{1}{n\times m}(A_{1}+A_{2}) \end{array}\right.$$
where $\nabla_{u}$ and $\nabla_{v}$ represent the first-order derivatives in the horizontal and vertical directions, respectively.

The third term primarily considers the statistical characteristics after denoising. The phase of interferometric images contains information about the shape and surface properties of targets. By considering statistical properties, it is possible to better maintain the consistency of phase in interferometric images, aiding in accurately reconstructing the phase information of targets. Previous studies have often used Kullback–Leibler (KL) divergence to assess statistical characteristics after denoising. However, KL divergence can only measure the difference in one distribution relative to another, and it cannot simultaneously evaluate the symmetric difference between two distributions. Therefore, this study uses Jensen-Shannon (JS) divergence, which addresses the asymmetry problem of KL divergence by considering the differences between both distributions simultaneously. Specifically, this is expressed as Equation (14):(14)$$L_{3}=\frac{1}{2}KL(P∥M)+\frac{1}{2}KL(Q∥M)$$
where *M* is the average distribution of *P* and *Q*, where *P* and *Q* represent the distributions of the model’s output phase map and the true phase map, respectively. KL(P∥M) denotes the Kullback–Leibler (*KL*) divergence from distribution *P* to *M*, defined as in Equation (15):(15)$$KL(P∥M)=\sum_{i}P(i)\cdot \log(\frac{P(i)}{M(i)})$$

The distributions of *P* and *Q* can be obtained from the histograms of the model’s output phase values and the true phase values, respectively, without requiring additional input parameters.

Finally, the overall loss can be expressed as Equation (16):(16)$$L=L_{1}+L_{2}+L_{3}$$

### 2.3. Evaluation Metrics

The evaluation metrics for simulation data include the Structural Similarity Index (SSIM) and MSE, which are used to assess the quality of the model’s output results. SSIM measures the similarity between the output result and the ideal result from a perceptual perspective. The calculation formula is as follows:

Equation (17) compares the brightness of two images, where $\mu_{x}$ and $\mu_{y}$ are their respective means, and $c_{1}$ is a constant.
(17)$$l(x,y)=\frac{2\mu_{x}\mu_{y}+c_{1}}{\mu_{x}^{2}+\mu_{y}^{2}+c_{1}}$$

Equation (18) compares the contrast of two images, where $\sigma_{x}$ and $\sigma_{y}$ are their respective variances, and $c_{2}$ is a constant.
(18)$$c(x,y)=\frac{2\sigma_{x}\sigma_{y}+c_{2}}{\sigma_{x}^{2}+\sigma_{y}^{2}+c_{2}}$$

Equation (19) compares the structure of two images, where $\sigma_{xy}$ is their covariance, and $c_{3}$ is a constant.
(19)$$s(x,y)=\frac{\sigma_{xy}+c_{3}}{\sigma_{x}+\sigma_{y}+c_{3}}$$

The final *SSIM*, as given in Equation (20), is the weighted product of the three factors mentioned above:(20)$$SSIM=l{\left(x,y\right)}^{\alpha }\cdot c{\left(x,y\right)}^{\beta }\cdot s{\left(x,y\right)}^{\gamma }$$
where *α*, *β*, and *γ* are parameters used to balance brightness, contrast, and structure comparisons, typically set to *α = β = γ* = 1. Ideally, a perfect filtering result should yield *SSIM* = 1.

*MSE* is a commonly used metric to assess the differences between two images, measuring the square of the average difference between predicted values and true values. It is calculated as shown in Equation (21):(21)$$MSE=\frac{1}{M\times N}\sum_{i=1}^{n}\sum_{j=1}^{m}{\left(I(i,j)-K(i,j)\right)}^{2}$$
where *I* and *K* are two images of size M×N, and the corresponding pixel values of each image are I(i,j) and K(i,j), respectively. A smaller value indicates that the two images are more similar or have smaller differences, ideally being zero.

For real data, due to the lack of real noiseless data, BM3D cannot recover the deformation phase, therefore it is challenging to directly compare the results with actual ground deformation for quantitative analysis. Therefore, we use the following quantitative and qualitative analysis methods to evaluate the performance of MOMFNet.
(1)Residual Map Analysis: Quantitative analysis is conducted using the difference between the noisy image and the denoised image, known as the residual noise map. An ideal filtering method should separate the signal from the noise, so the signal remaining in the residual noise map reflects the imperfection of the filtering method. In the residual noise map, the more phase trends are highlighted, the more signal is suppressed, indicating a better filtering effect.(2)Deformation Center Phase Variation Analysis: By observing the plot of phase changes across the deformation center, the oscillation of the filtered phase trend can be analyzed. Greater oscillation indicates more noise interference in the image, while lower oscillation suggests less noise interference.(3)Residual Count Analysis: The total number of pixels in the residual map is calculated to quantitatively assess the denoising effect. A smaller residual count indicates better denoising performance.

Through these three analysis methods, we can qualitatively and quantitatively evaluate the denoising effectiveness of MOMFNet and compare it with other denoising methods.

## 3. Results

### 3.1. Simulation Experiment Results Analysis

MOMFNet’s training and test sets consist of 8000 and 2000, 180 × 180-pixel floating-point arrays, respectively. The training iterations (Epochs) were set to 200, with a training time of approximately 49.2 h. To more clearly validate the model’s reliability and scalability on simulated data, this study increased the noise intensity compared to previous experiments. Specifically, π, 2π, and 3π noise intensities were added to samples one, two, and three, respectively, each using different terrain conditions. The proposed method was compared against DnCNN [53], InSAR-BM3D [25], and Goldstein [16].

As shown in Figure 3, when the noise level is π, the proposed MOMFNet performs very close to noise-free interferograms in terms of detail and preservation of mining area edges. DnCNN is capable of reproducing interferogram shapes with low noise but still exhibits some noise. InSAR-BM3D generates smooth and clear edges and overall structures but struggles to preserve minor variations in deformation edges. The Goldstein method presents disturbed stripes with significant phase impact on the image. When the noise reaches 2π, MOMFNet still ensures low noise and relatively smooth deformation edges. DnCNN shows minor disturbances in deformation edges despite its overall good structure. InSAR-BM3D exhibits distortions in the deformation center area, though it performs well in overall noise reduction. Goldstein filtering struggles significantly to recover the overall deformation phase of the interferogram. When facing challenging noise levels of 3π, MOMFNet achieves good overall smoothing effects with minor edge errors, but there are some deformations in the deformation center due to the high noise levels. DnCNN exhibits disturbances in areas with steep deformation gradients and minor blurring at edges, yet it generally provides smooth noise reduction.

InSAR-BM3D cannot recover the deformation phase in the deformation center area effectively, resulting in moderate overall noise reduction. Goldstein filtering faces even greater challenges in restoring the overall deformation phase of the interferogram. Therefore, from a visual interpretation perspective, MOMFNet, proposed in this study for denoising, demonstrates relatively good performance in smoothing deformation edges and preserving the phase under high noise conditions.

The average processing time for filtering a single image using different methods is shown in Table 1. MOMFNet takes only 0.16 s to process a single sample, making it faster than all traditional unfolding methods. Compared to deep learning methods, DnCNN has a relatively lower time consumption due to its simpler network architecture, which results in higher computational efficiency than MOMFNet. However, its filtering performance on the test set is relatively worse.

To further analyze the performance of different methods, Table 2 presents the quantitative analysis results based on the MSE and SSIM metrics, evaluated across the entire test dataset. In Table 1, MSE1, MSE2, and MSE3 represent the Root Mean Square Errors for samples one, two, and three, respectively. In Table 1, SSIM1, SSIM2, and SSIM3 represent the Structural Similarity Indices for samples one, two, and three, respectively. The best values are highlighted in bold. As shown in Table 1, MOMFNet achieves relatively good values in both metrics, followed by DnCNN, InSAR-BM3D, and Goldstein. It is evident that these quantitative metrics further reflect the results of visual interpretation analysis.

### 3.2. Real-SAR Images Experiment Results Analysis

To further test the algorithm’s performance, we applied MOMFNet to two coal mining areas in Yanzhou, Jining City, Shandong Province, China. The study area is centered approximately at latitude 35°03′01″ N, longitude 115°56′18″ E, with an elevation of about 47 m. This section compares the qualitative and quantitative aspects of real Sentinel-1 data.

The results of Case 1 are shown in Figure 4. The first row shows the noisy interferogram and the interferograms filtered by various methods. It can be observed that all methods exhibit good and noticeable noise suppression effects. However, upon closer inspection, significant differences can be identified. The Goldstein filtering method shows a noticeable distortion in the deformation trend after denoising, with a relatively poor noise suppression effect. While the InSAR-BM3D method achieves relatively better denoising, resulting in a smoother area around the deformation, it fails to reveal the deformation trend at the deformation center. The DnCNN filtering method restores some deformation trends and exhibits overall good denoising performance. However, there is some fluctuation in the interferometric fringes at the deformation center, which is due to the residual noise at the deformation edge affecting the generated phase. The large deformation gradient in the mining area, mixed with significant noise, causes the phase corresponding to the edge to perform poorly in DnCNN. The proposed MOMFNet effectively overcomes these issues, removing most of the noise without causing distortion in the central deformation area due to a large deformation gradient, and better preserves the phase variation.

The second row of Figure 4 shows the coherence map and the residual maps of each filtering method. As shown in the second row of Figure 4, the selected study area has poor coherence, and all residual maps display a certain degree of randomness, confirming the good filtering process of each method. However, compared to other methods, MOMFNet does not exhibit significant coherent patterns. Additionally, as shown in the results of the first row, all methods show some degree of noise residual in the central deformation area.

Figure 5 shows the noisy interferogram, the interferograms filtered by each method, and the phase variation trends across the deformation center. It can be observed that the phase trend oscillation is relatively smoother for MOMFNet compared to other methods, indirectly revealing that the results of MOMFNet are relatively better.

The results of Case 2 are shown in Figure 6, where we compare the denoising effects of various methods in Case 2. The first row’s first image shows the original interferogram, which contains a significant amount of phase noise. This noise makes it difficult to recognize phase changes, affecting the quality and interpretation of the interferogram. The Goldstein filtering method still leaves some noise. The InSAR-BM3D and DnCNN methods perform well in noise reduction, significantly reducing noise and making the fringes and phase jumps clearer. The InSAR-BM3D method is relatively smoother, but some noise remains, especially at the deformation edges. The MOMFNet method shows outstanding performance in noise reduction, effectively suppressing phase noise while preserving phase details and edge information in the interferogram. The second row shows the residual noise maps of each method. As seen in the second row, each method more clearly shows the trend of phase at the deformation center, while some useful information is lost during noise reduction. MOMFNet’s noise appears more random, indicating better denoising performance.

As shown in Figure 7, we compared the linear results of the noisy and denoised interferograms across the deformation center. Other methods appear relatively smoother compared to the original noisy image and have lower oscillation frequency, while MOMFNet shows better performance. Therefore, the MOMFNet method demonstrates stronger robustness in handling large gradients and random noise, with more significant denoising effects.

Finally, to further quantitatively validate the results, the residual total for each image was calculated, as shown in Table 3. MOMFNet has a smaller residual total compared to other methods. Generally, the smaller the residual total, the closer the processed image is to the original image, indicating better processing performance.

## 4. Discussion

### 4.1. Ablation Study of the Multi-Objective Weighted Network

This study proposes a Multi-Objective Multi-Kernel Feature Extraction-Based Deep Learning Method for InSAR Phase Filtering, where the introduced loss function adopts a multi-objective strategy. During filtering, the network simultaneously considers spatial, edge, and statistical characteristics while incorporating weighted residual blocks and added a multi-convolutional kernel feature extraction module. This allows the network to adaptively adjust the importance of features, enhancing the effectiveness of the model. We conducted an ablation study to evaluate the performance of the proposed network. This experiment randomly selects 2500 simulation data, of which 2000 are the training set and 500 are the testing set, and sets 100 epochs of results for discussion.

As shown in Figure 8, before the dashed line “a”, this stage corresponds to the early phase of training, where the loss values of all models decrease rapidly, indicating that the models are learning and adapting to the data features. Among them, DnCNN+WRB exhibits noticeable fluctuations in this phase, suggesting that its weight updates may be unstable during early training. MOMFNet, on the other hand, shows the smoothest decline in the loss curve, demonstrating strong convergence and stability. Between the dashed lines “a” and “b”, the loss values continue to decrease, but the rate of decline slows down, indicating that the models are approaching their optimal solutions. During this phase, MOMFNet maintains a steady decline and consistently outperforms the other models, proving its superior optimization capabilities. DnCNN+MF and DnCNN+MO also demonstrate good learning performance but have slightly higher loss values compared to MOMFNet. Meanwhile, DnCNN+WRB and DnCNN still exhibit significant fluctuations, showing less stability in convergence compared to the other models. After the dashed line “b”, the loss values tend to stabilize, indicating that the models have largely completed their learning and are approaching convergence. During this phase, MOMFNet continues to achieve the lowest loss values, showing its superiority in the final stage. DnCNN+MF and DnCNN+MO, while inferior to MOMFNet, still exhibit relatively strong optimization performance. However, the training processes of DnCNN+WRB and DnCNN remain unstable, and their final performance is significantly lower than the other methods. This analysis highlights that MOMFNet’s multi-objective multi-kernel feature extraction strategy effectively enhances convergence and improves performance.

To further quantitatively analyze the results, as shown in Table 4, we evaluate the models using the root mean square error (RMSE) on the validation set, the mean absolute error (MAE) on the validation set, and the RMSE on the test set. The metrics also indicate that method L5 is relatively superior.

### 4.2. Reliability of Phase Trend in Validating Results on Real Data

In the field of InSAR denoising, verifying the effectiveness of denoising on real data has always been a challenging task. This difficulty arises because it is impossible to accurately obtain noise-free interferograms for real data, rendering many analytical metrics inapplicable. We will now discuss the reliability of using phase trend maps to validate the denoising results on real data.

As shown in Figure 9, we simulated noise-free interferograms, interferograms with noise levels of π, and interferograms with noise levels of 3π, and extracted their phase values along the deformation center to generate deformation trends. It is evident that the phase trend corresponding to the noise-free interferogram is very smooth and clearly reflects the deformation subsidence trend. In contrast, the phase trend corresponding to the interferogram with random noise of π can only roughly indicate the deformation subsidence trend, with larger amplitude and frequency fluctuations, and some areas exhibit abrupt changes, making it difficult to discern trends in regions of intense deformation. For the phase trend corresponding to the interferogram with random noise of 3π, the entire line chart appears random, making it impossible to recognize the overall deformation subsidence trend, and the amplitude and frequency are significantly higher. Therefore, the noise level shows a strong correlation with the smoothness of the phase deformation trend along the deformation center, providing a basis for assessing the quality of denoising results in real experiments.

## 5. Conclusions

To suppress phase noise and improve the quality of interferograms, this study proposes MOMFNet: a deep learning approach for insar phase filtering based on Multi-Objective Multi-Kernel Feature Extraction—MOMFNet. The method introduces a multi-objective loss function that considers both the spatial and statistical characteristics of interferograms and extracts information of different scales through multiple convolutional kernels. It also incorporates learnable weights in the residual network to achieve adaptive adjustment and enhance the model’s flexibility. To ensure the authenticity and diversity of training data, the study simulated atmospheric turbulence phases using randomly distorted 2D Gaussian surfaces and fractal Perlin noise, supplemented with negative Gaussian noise. To validate the effectiveness of the proposed method, both simulated and real-world data were quantitatively and qualitatively evaluated, and the results were compared with existing advanced phase denoising algorithms. The findings demonstrate that MOMFNet excels in noise suppression and edge preservation, effectively maintaining phase variations, fringe edges, and phase jumps. Moreover, it exhibits robust performance across different noise levels, confirming the stability and reliability of MOMFNet.

This research enriches the development of InSAR denoising technology by proposing a novel denoising approach. However, the study primarily relies on simulated data, which, despite efforts to approximate real-world scenarios, still exhibits some gaps. Additionally, the complexity and computational cost of the model restrict its application in larger and more complex environments.

## Figures and Tables

**Figure 1 sensors-24-07821-f001:**
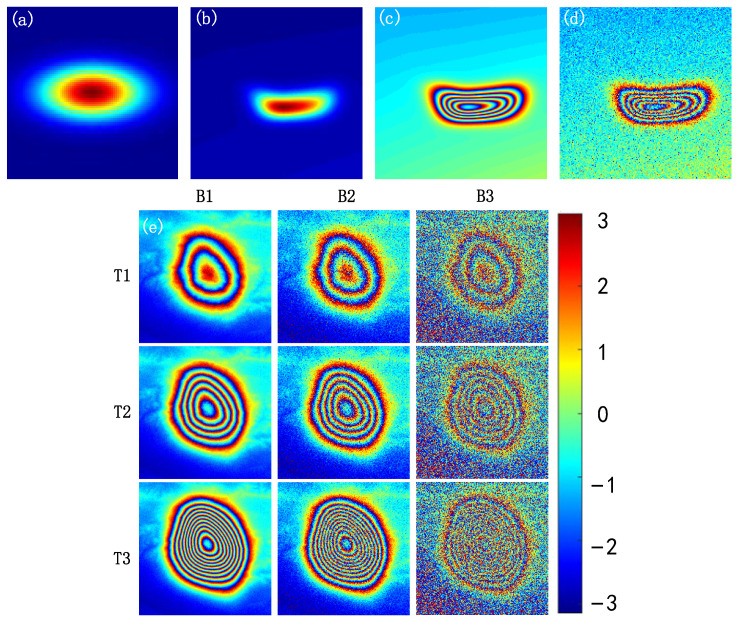
Subfigure (**a**) shows the deformation phase map generated by a 2D Gaussian ellipse; subfigure (**b**) presents the deformation phase map generated by a distorted 2D Gaussian ellipse; subfigure (**c**) displays the interferogram; subfigure (**d**) illustrates the interferogram with noise; Generation of dataset noise: subfigure (**e**) shows, from left to right, the noise-free interferogram, the interferogram with noise level π, and the interferogram with noise level 2π. From top to bottom, the deformation gradient of the interferogram continuously increases.

**Figure 2 sensors-24-07821-f002:**
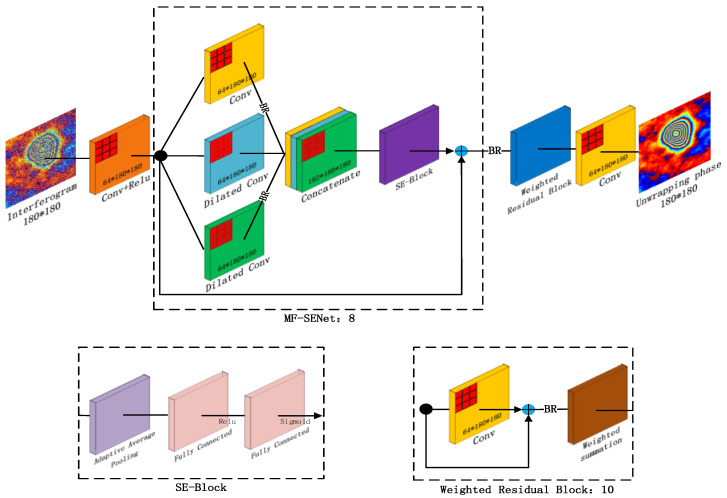
Design scheme of the network structure.

**Figure 3 sensors-24-07821-f003:**
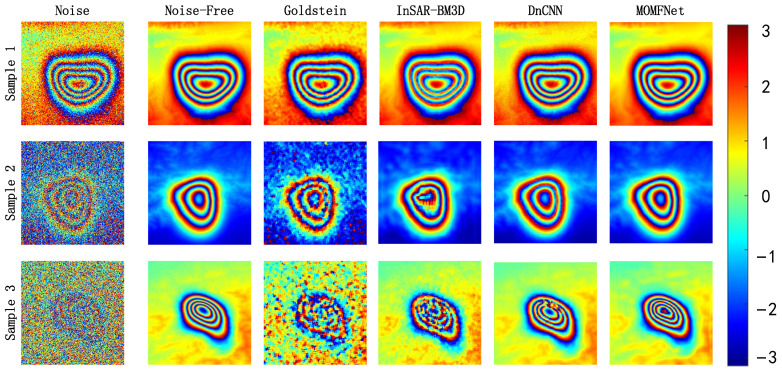
The results on simulated interferograms with different deformation magnitudes and noise levels. Each row shows the interferograms of samples one to three with noise levels π, 2π, and 3π, respectively, representing different terrains. Each column represents the noise-free interferogram, Goldstein, InSAR-BM3D, DnCNN, and MOMFNet denoising results, respectively.

**Figure 4 sensors-24-07821-f004:**
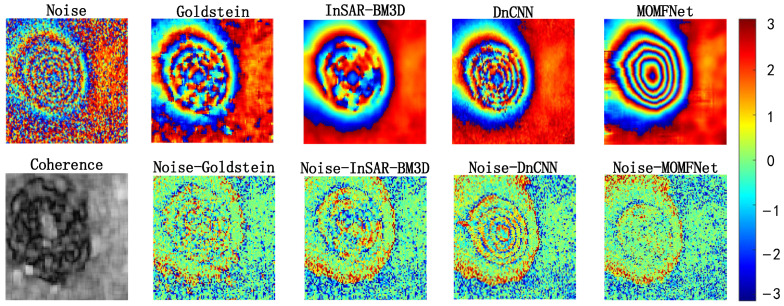
Experimental results for case one. The first row shows the noisy interferogram and the denoised image, respectively. The second row shows the coherence map and the corresponding residual noise maps.

**Figure 5 sensors-24-07821-f005:**
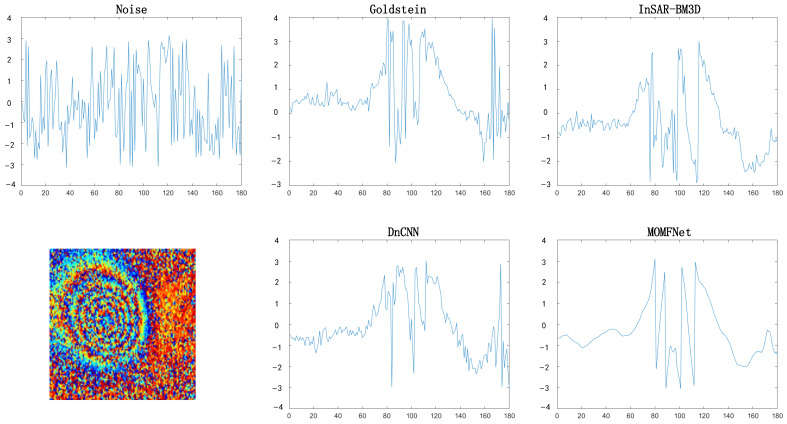
The phase variation trends through the deformation center for case one: comparing interferograms with noise and denoised interferograms.

**Figure 6 sensors-24-07821-f006:**
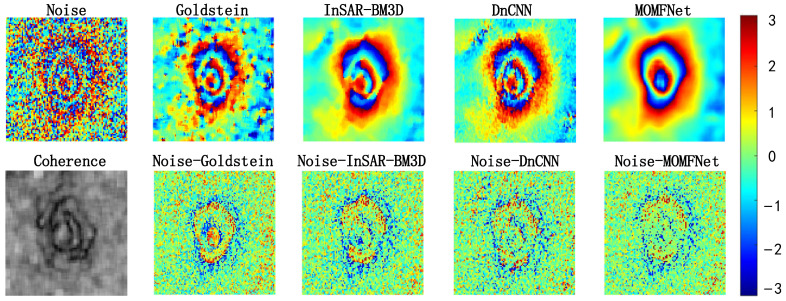
Experimental results for case two. The first row shows the noisy interferogram and the denoised image, respectively. The second row shows the coherence map and the corresponding residual noise maps.

**Figure 7 sensors-24-07821-f007:**
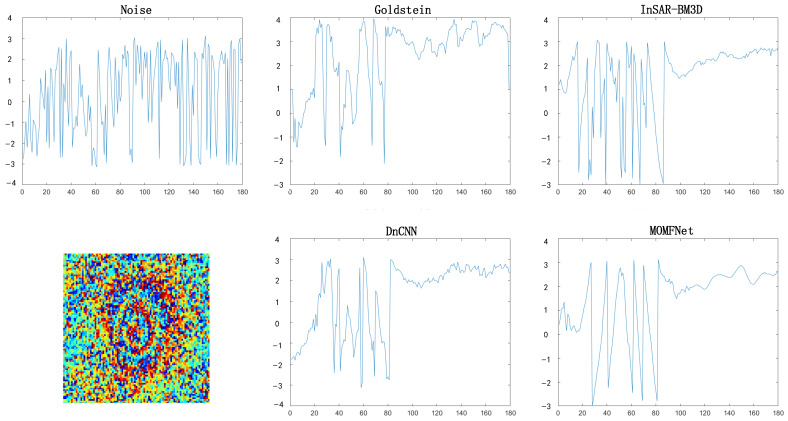
The phase variation trends through the deformation center for case two: comparing interferograms with noise and denoised interferograms.

**Figure 8 sensors-24-07821-f008:**
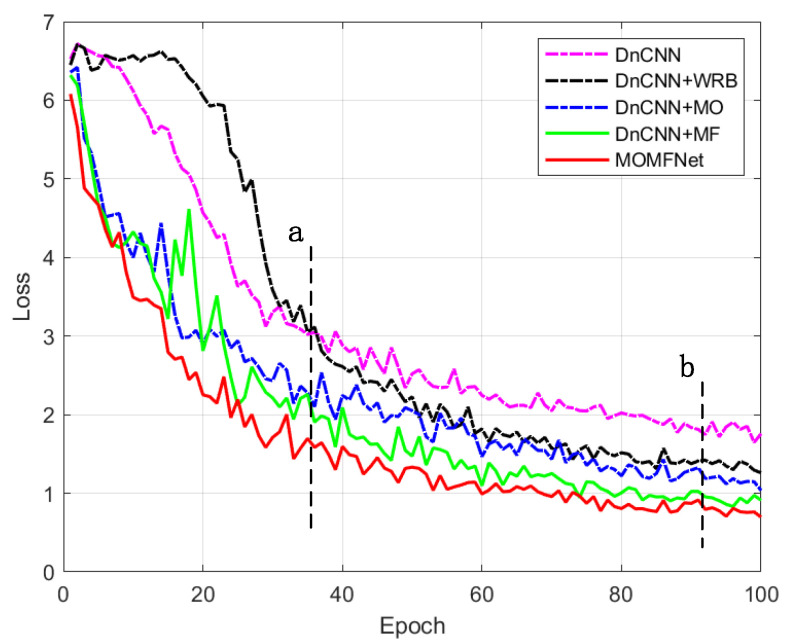
Trend of loss value changes induced by different methods.

**Figure 9 sensors-24-07821-f009:**
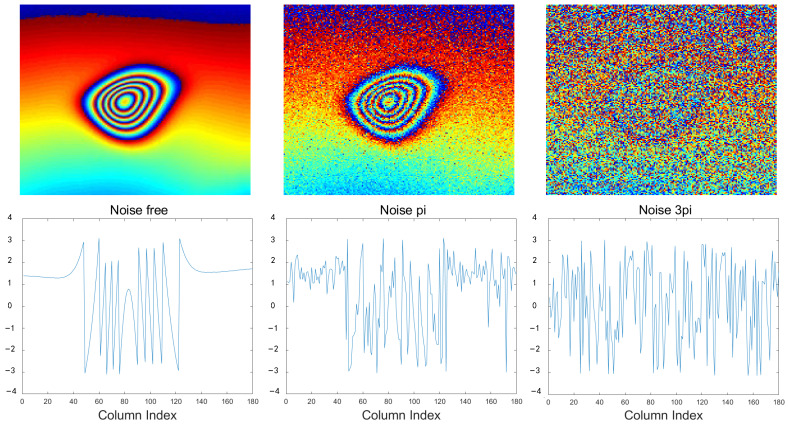
The phase variation trends through the deformation center for simulated Data: The first row shows the interferograms with no noise, with noise of π, and with noise of 3π. The second row displays the corresponding phase change maps through the deformation center for these interferograms.

**Table 1 sensors-24-07821-t001:** The filtering time of different methods.

Method	Time
Goldstein	0.24
InSAR-BM3D	0.46
DnCNN	0.10
MOMFNet	0.16

**Table 2 sensors-24-07821-t002:** Comparison of different methods of MSE and SSIM.

Method	MSE1	MSE2	MSE3	SSIM1	SSIM2	SSIM3
Goldstein	3.4724	26.891	64.5238	0.1059	0.0338	0.0059
InSAR-BM3D	1.4329	2.6734	12.324	0.3382	0.1673	0.0982
DnCNN	0.9112	1.1321	2.6358	0.4367	0.3254	0.1832
MOMFNet	0.5517	0.6853	1.1564	0.7337	0.6853	0.5342

**Table 3 sensors-24-07821-t003:** The total number of residues.

Noise	Cass 1 50,124	Cass 2 17,398
Goldstein	34,784	11,235
InSAR-BM3D	28,675	6496
DnCNN	30,714	8798
MOMFNet	**12,065**	**2463**

**Table 4 sensors-24-07821-t004:** Numerical evaluation of different methods.

Method	*RMSE* (Val)	*MAE* (Val)	*RMSE* (Test)
L1	2.289	1.915	2.981
L2	1.592	1.038	2.125
L3	1.349	0.882	1.878
L4	1.416	0.773	1.851
L5	**1.294**	**0.716**	**1.749**

## Data Availability

Sentinel data were made available by ESA in the Copernicus project through the Open Access Hub portal (https://scihub.copernicus.eu/dhus/#/home accessed on 20 March 2023).

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
