# Peer review of "MOMFNet: A Deep Learning Approach for InSAR Phase Filtering Based on Multi-Objective Multi-Kernel Feature Extraction"

_sensors, 2024, doi:10.3390/s24237821_

Round 1

Reviewer 1 Report

Comments and Suggestions for Authors

This paper proposed a deep learning approach for InSAR phase filtering based on multi-objective multi-kernel feature extraction. It should be undergone a major revision and solve the following comments. MF-SENet is similar to dilated convolution module (dc-module) in hyper-light.

Deep learning network for high-accurate and high-speed ship detection from synthetic aperture radar imagery, thus compare them in the texts.

Please clarify the novelty of the network.

Synthetic aperture radar applications can also consider detection works such as high-speed ship detection in sar images based on a grid convolutional neural network and depthwise separable convolution neural network for high-speed sar ship detection.

Some InSAR phase filtering methods should be compared including cfar-guided dual-polarization fusion framework so as to sparse-model-driven network for efficient and high-accuracy insar phase filtering.

I want to know if segmentation models can be used for this work likewise scale-aware dimension-wise attention network for small ship instance segmentation, dual-polarization information-guided network for sar ship classification.

Finally, some English should be improved.

Reviewer 2 Report

Comments and Suggestions for Authors

This manuscript proposes NOMFnet for InSAR Phase Filtering. There exists 6 major issues . Comments are listed as follows.

Major issue 1: Most references are published before 5 years ago in the part of introduction. Authors should cite newer scholar papers in introduction. The rate of newer scholar papers in introduction should be more 30%.

Major issue 2: In the processing of InSAR, the data is usually utilized after phase unwrapping. Authors should present research results on phase unwrapped data rather than phase wrapped data.

Major issue 3:Since the phase is wrapped in this manuscript, the noise of pi should equal to the noise of 3pi due to period of 2pi. How do authors add noise of pi and noise of 3pi as presented in Figure 9?

Major issue 4: Authors should also present time cost of the deep learning methods and traditional methods.

Major issue 5: Authors should summarize the main contributions in the part of introduction.

Major issue 6: Authors should reorganize description in 2.1.1 in order to describe the structure of MOMFNet more clearly.

Reviewer 3 Report

Comments and Suggestions for Authors

The author presented a Deep Learning Approach for InSAR Phase Filtering Based on Multi-Objective Multi-Kernel Feature Extraction. The manuscript is well written backed with indepth literature review and analysis. Although there is no major issue in the manuscript, there are still some changes are required to further imporve the quality.
The termonologies like MOMFNet must be defined first time when used in abstract or introduction and then used in abreivated form.
A lot of useless references are utilized, authors are requested to narrowdoen the references and only cite the related state of the art.
The equations must be ciyte properly with there references.
Figure 2 must be written in sub figure (a) (b) to provide a clear information.
The quality of Figure 8 must be improved.

Comments on the Quality of English Language

The English could be improved to more clearly express the research.

Round 2

Reviewer 1 Report

Comments and Suggestions for Authors

no more comment 

Reviewer 2 Report

Comments and Suggestions for Authors

Authors have replied to all issues that reviewer concerned.